# Strategies for Altering Delivery Technologies to Optimize CAR Therapy

**DOI:** 10.3390/ijms26073206

**Published:** 2025-03-30

**Authors:** Lili Cao, Yingying Liu, Guimei Lin

**Affiliations:** 1Student Counseling Center, Shandong University, Jinan 250012, China; caolili@sdu.edu.cn; 2School of Pharmaceutical Science, Shandong University, Jinan 250012, China; 15763554145@163.com

**Keywords:** CAR-T cells, solid tumor, immunotherapy, delivery vector

## Abstract

Chimeric antigen receptor (CAR) T-cell therapy has been proven to be an effective strategy for the treatment of hematological malignancies. At present, how to prepare CAR-T cells efficiently, quickly, and safely is one of the urgent problems to be solved. The durability and activity of engineered T cells in solid tumors need to be further improved, and the strategy of T cells penetrating the tumor microenvironment also needs to be improved. In addition, although the problems mainly caused by T-cell biology are being solved, the manufacturing mode and process still need to be improved to ensure that CAR-T cell therapy can be widely used. This paper summarizes some strategies that can improve the efficacy of CAR-T cells.

## 1. Introduction

Cell therapy has attracted much attention because of its remarkable efficacy and good prognosis in hematological tumors [1,2,3]. At present, researchers begin to introduce the T-cell receptor (TCR) or CAR gene that recognizes tumor antigens into lymphocytes through gene modification to make them become TCR gene-modified T lymphocytes (TCR-T) or CAR-T cells, so that they have the ability to target tumor antigen and kill tumor cells [4,5]. CAR technology has been widely concerned and developed at a high speed and has achieved breakthrough success in the treatment of blood tumors [6,7]. CAR-T therapy, namely chimeric antigen receptor T-cell immunotherapy [8], allows patients’ T cells to express chimeric antigen receptors through gene-transformation technology, making the targeting, killing, and persistence of effector T cells higher than those of immune cells used routinely, which can overcome tumor local immune suppression microenvironment and break host immune tolerance. The specific process of this therapy is to collect human autoimmune cells, increase their number, and enhance targeted killing function after in vitro culture, and then reinfuse them to the human body to kill pathogens, cancer cells, and mutant cells in the blood and tissues, activate and enhance the immune capacity of the body, and play a therapeutic and health care role. CAR-T immunotherapy can be used to treat blood tumors such as acute leukemia and non-Hodgkin’s lymphoma, and it can also be modified to treat solid tumors [9,10], autoimmune diseases [11], HIV infection, heart disease, and other diseases [12,13,14,15], so it has a broad application space. At present, there are more than 800 CAR-T cell therapy drugs developed in the world, eight of which have been approved for marketing, including six CAR-T therapies approved by the FDA, and two CAR-T products—Aquilence injection and Ricky Orensei injection—have also been listed in China, but most of them are related to the treatment of blood tumors, and no substantial breakthrough has been made in the treatment of solid tumors.

There are many limitations of CAR-T cell therapy to be solved, including limited persistence, life-threatening CAR-T cell-related toxicity, immunosuppressive microenvironment, poor migration and tumor invasion, and antigen escape [16]. Therefore, strategies need to be taken to make CAR-T cells more effective in efficacy and improve safety. At present, CAR-T cells are mainly produced in vitro and in vivo. Preparation in vitro is a common method, but directly producing CAR-T cells in patients can be said to be the most technically challenging and clinically useful method. Preparation of CAR in vivo can make up for the high cost of preparation in vitro. In this review, based on the latest research progress for the past 10 years, we mainly discussed some in vitro and in vivo delivery strategies. And some delivery methods can improve the efficacy of CAR therapy to improve the treatment durability, and proposed solutions to some problems faced by CAR therapy [17].

## 2. Main

CAR-T cell therapy is a tumor therapy that has made breakthrough progress internationally, but its main disadvantage is that it is very expensive, with the price of each dose up to hundreds of thousands of dollars. The main reason is that the treatment cost of CAR therapy is very high, and one of the reasons for the high cost is that the manufacturing process is complex, time-consuming, and needs to be customized for patients [18]. In addition, the limited persistence is also a big challenge for CAR-T therapy, because in the tumor microenvironment, there are many soluble factors that can promote tumorigenesis and inhibit the function of CAR-T cells [19]. These cytokines have a negative impact on the survival, invasion, and activity of T cells. In fact, one of the signs of cancer is that excessive inhibitory determinants in TME lead to impaired T-cell function, which leads to depletion of T cells [20]. Therefore, in order to better solve the above problems, we summarized some in vitro and in vivo delivery technologies that can improve the efficiency of CAR therapy.

Traditionally, the technology of producing CAR-T cells in vitro is generally used, that is, the edited T cells are transfused back into the patient’s body through intravenous infusion and then transported to the tumor site through blood circulation to play a therapeutic role. In this process, we can start from the basic structure of CAR-T cells, optimize their surface structure, and improve their ability to specifically combine with target cells, to enhance the anti-tumor efficacy; After the preparation of CAR-T cells, we can also change the way of administration, such as local injection, to help improve the persistence of cells in the tumor site, and improve the problem of CAR-T infiltration and transport.

Secondly, CAR-T cells can be produced not only in vitro, but also directly in vivo. In this article, we will introduce some techniques for producing CAR-T cells in vivo. In vivo gene therapy injects CAR encoded vectors into the bloodstream once, enabling T cells to decode new genes and generate CAR in a day or two. This simple treatment scheme may make CAR-T cell therapy more widely applicable to patients.

Finally, CAR-T cells combined with immune checkpoint inhibition therapy is also an effective strategy for the treatment of malignant tumors, specific ways to improve the survival rate of T cells by combining cytokine for synergistic therapy. In clinical treatment, it is necessary to determine whether combined treatment is required and select the appropriate combined treatment scheme according to the specific situation of the patient [21].

### 2.1. In Vitro Production Mode

At present, the commonly used clinical therapy is to produce CAR-T cells in vitro and then transfuse them back [22]. The whole process of producing CAR-T cells in vitro is simply to separate lymphocyte T cells from patients’ peripheral blood, transfer the CAR gene encoding tumor antigen specificity into T cells through gene carriers, and generate CAR-T cells [23]. After a certain number of CAR-T cells are amplified in vitro, they are transfused back to patients via vein and transferred to tumor sites, complete CAR-mediated target antigen tumor recognition, and precisely kill tumor cells. In this process, we can start from the basic structure of CAR-T cells, optimize their surface structure, and improve their specific binding ability with target cells, so as to enhance the anti-tumor effect. After the preparation of CAR-T cells, we can also change the way of administration, such as local injection, to help improve the persistence of cells in the tumor site, and improve the problem of CAR-T infiltration and transport. Next, we will discuss several ways to improve the efficiency of CAR therapy in the above-mentioned process of producing CAR-T cells in vitro.

#### 2.1.1. CAR Structure Optimization

The design of CAR molecules is the key link of CAR-T cell therapy [24]. Before delivering CAR-T cells back to patients, the design of the CAR-T structure can be improved and optimized to improve the durability of CAR-T cells and enhance the anti-tumor effect. CAR is a modular synthetic receptor [25]. Its basic structure includes a tumor-associated antigen (TAA) binding region (usually from the scFV segment of the monoclonal antibody antigen binding region), an extracellular hinge area, a transmembrane region, and an intracellular immune receptor tyrosine-based activation motif (ITAM) [26]. The tumor-associated-antigen TAA only exists on the surface of tumor cells and does not express in any normal tissues, and different TAA can be used as the target antigen of CAR-T cells and play a targeted role in CAR-T.

The selection of target antigen is a key determinant for the specificity and effectiveness of CAR and the safety of genetically modified T cells [27,28]. Therefore, the selection of appropriate target proteins is an important aspect of CAR-T cell immunotherapy in the clinical research and application of cancer. At present, the clinical development of antibodies is mainly aimed at targeting tumor-related antigens (TAA, such as CD19, CD20, CD22, CD30, CD33, BCMA, and other targets) [29,30]. Among them, CD19, as a target, is frequently expressed in B-cell leukemia and lymphoma [31], and the degree and level of expression are much higher than other potential targets, such as CD20 and CD22 [32]. Therefore, CAR-T targeting CD19 is the most successful and widely used cell therapy. In recent years, monoclonal antibody, double antibody, ADC, CAR-T drugs targeting CD19 have been marketed, providing a variety of treatment methods for patients with advanced B-cell lymphoma [33]. However, in addition to CD19, preclinical and clinical targets for malignant B-cell tumors also include ROR1, CD37 and multiple myeloma, kappa light chain, CD138, SLAMF7, CD38, and integrin β7 [34]. It is also of great significance for tumor therapy to carry out CAR therapy research on these new targets.

Another important aspect of CAR-T cell structure optimization is the introduction of costimulatory molecules. With the continuous advancement of scientific and technological research, CARs technology continues to innovate and has developed to the fifth generation. The first-generation CARs [35] contain extracellular single-chain variable fragment scFv, transmembrane region and single intracellular activation signal CD3ζ or FcεRIγ, but without costimulatory molecules, the first generation CAR-T cells can only cause transient T-cell proliferation and less cytokine secretion. It ultimately failed to achieve the expected therapeutic efficacy in clinical application. Carl H. June’s team from the University of Pennsylvania designed the second-generation CAR-T with CD19 as the target and 4-1BB as the costimulatory domain [36]. The second-generation CARs introduced a costimulatory molecule on the basis of the first generation, improving the tumor-killing effect [37]. The third-generation CARs are similar to the second-generation CARs. They carry multiple costimulatory factors, such as CD28, CD134 (OX40), and CD137 (4-1BB). The costimulatory molecules can activate JNK, ERK, NF in T cells- κ B and other signal pathways can prolong the anti-tumor activity, proliferation activity, survival cycle and cytokines (such as IL-2 and TNF) of T cells-α and IFN-γ). The secretion and other aspects were significantly improved. In two experiments, CAR-T cells transduced by the third-generation CARs were used in leukemia and non-Hodgkin’s lymphoma (NHL), respectively, and the results showed superior amplification and longer persistence [15,38].

In addition to introducing costimulatory factors, in order to enhance the effective cytotoxicity of CAR-T to target tumor cells, the tumor microenvironment can be regulated by simultaneously secreting cytokines. The fourth-generation CARs are designed by adding new functions beyond T-cell activation signals, including secreting a variety of cytokines and additional antibodies, which can regulate the immune microenvironment in solid-tumor tissues. In structure, the fourth-generation CARs added selective markers and coded CARs amplification promoters based on the third-generation and added suicide genes or controllable suicide genes (such as some drug-sensitive genes) to control the survival time of CAR-T cells in vivo. At the same time, when CARs combine with targeted antigens, they can release the products of modified genes into tumor tissues. The local transfer of immune regulatory molecules, such as pro-inflammatory molecules, cytokines, and enzymes, make the tumor microenvironment of T-cell immunosuppression transform into the immune-tolerant tumor microenvironment, thus increasing the infiltration of T cells in tumor tissue and achieving the effect of enhancing the killing of solid tumors [39].

Generally, CAR-T therapy is a highly personalized therapy that is engineered in vitro by patients’ autologous T cells and then reinfused. The treatment process is complex, and the cost is high. Therefore, when developing the fifth-generation CARs technology, the universal CAR [40] uses two “third-party” systems, BBIR CAR (biotin-binding immune receptor) or SUPRA CAR (split, universal, programmable), to split the extracellular antigen-targeting domain and T-cell signal units. In order to endow CAR-T cells with the ability to recognize multiple antigens, this kind of universal CAR-T also has higher technical barriers and higher requirements for safety. At present, the research on universal CAR-T in China is still in its infancy.

To sum up, for the optimization of CAR structure, the first- to third-generation CARs do not change the intracellular and extracellular infrastructure of chimeric antigen receptors, but only replace and improve the basic components of CAR. That is, the second and third generation introduced costimulatory factors, to improve the killing power of T cells and achieve longer durability. The fourth generation of CARs builds new structures or introduces new components based on changing the infrastructure. For example, based on the original structure, the function of secreting the immune factor interleukin-12 (IL-12) can be increased [39], and suicide genes can be introduced. Compared with the second- and third-generation CAR-T cell therapy, the fourth-generation CAR-T cell therapy shows better therapeutic effect. The fifth-generation CAR-T technology will break through the individual limitation and can be used universally among different individuals for large-scale production and treatment. To summarize, the iterative schematic of the optimization of the chimeric antigen receptor structures from the first to the fifth generation is shown in Figure 1.

In conclusion, the progress of CAR-T cell therapy cannot be separated from multiple generations of technological innovation [41]. The first generation of CAR-T has a basic structure but only integrates the CD3ζ signaling domain, which has the disadvantages of insufficient T-cell expansion and poor persistence when used, resulting in a relatively short persistence in the patient’s body. Second-generation CAR-T introduced [42], for example, the 4-1BB co-stimulatory domain, which significantly enhanced T-cell survival time and became the first FDA-approved cell therapy milestone, but faced side effects such as cytokine storm. To further improve therapeutic efficacy, third-generation CAR-T superimposed a second co-stimulatory signaling domain that could complement each other and increase effectiveness. Ramos et al. published an in vivo study of third- versus second-generation CD19-specific CAR-T cells in B-cell non-Hodgkin’s lymphoma (NCT01853631) [38]. The study showed that third-generation CAR-T cells expanded better (up to 40-fold) and lasted longer compared to second-generation. However, not all patients respond to this therapy. Fourth-generation CAR-T is genetically engineered to remodel the tumor microenvironment by secreting IL-12, among other things, and studies have shown that TRUCK CAR-T cells have shown complete remission rates of up to 75% in Hodgkin’s lymphoma and up to 100% in multiple myeloma [43,44]. To further reduce the cost of manufacturing time and increase versatility, a fifth generation of universal CAR-T has been created, and *The Lancet* has published the results of the NCT02808442 and NCT02746952 studies, which examined TALEN-edited CAR-T cells in patients with B-cell ALL. They showed a CR or CRI of 67% and an overall survival (OS) of 55%, with challenges such as safety optimization, adaptation to the solid tumor microenvironment, and cost controllability still to be addressed [45].

#### 2.1.2. Intravenous Delivery

After optimizing the molecular structure of the CAR gene, we often inject exogenous engineered and improved active immune cells (CAR-T cells) into the vein to reach the tumor site, maintain or even strengthen the immune system function of the body, and achieve the purpose of killing tumor cells. CAR-T cells are mainly transmitted by intravenous infusion, which is effective for the treatment of blood cancer because T cells can easily find and eradicate cancer cells [46].

Intravenous reinfusion has an obvious effect on hematological tumors, such as recurrent B-cell acute lymphoblastic leukemia [47,48]. KTE-X19 is a CAR-T cell product targeting CD19 [49]. It shows a high tolerance rate and controllable safety in adult patients with relapsed/refractory B-cell acute lymphoblastic leukemia (R/R B-ALL). After receiving a single infusion of KTE-X19, 71% of patients reached CR/CRI (CR in complete remission or CR in incomplete recovery of blood cell count), and 31% of patients continued to remission at the end of data. Craig M. Rive et al. showed that direct intravenous infusion of conventional VSV-g lentiviral particles carrying the CD19 CAR gene can transfer T cells, and then completely ablate endogenous B cells in wild-type mice [50]. Intravenous infusion of CAR-T cells for chronic lymphocytic leukemia, lymphoma including non-Hodgkin’s lymphoma, r/r follicular lymphoma (FL), Diffuse large B-cell lymphoma, and multiple myeloma have an obvious effect, but the effect on solid tumors is not obvious. The main reason may be that solid tumors and blood tumor cells are different in dispersion. Solid tumors generally present a cluster state. In addition, there are abundant related fibroblasts (CAF) and blood vessels, which form a physical barrier on the surface. It is difficult for drugs to effectively act on the target site. On the other hand, some solid tumors may inhibit the secretion of certain chemokines, and the interaction between chemokines and their receptors will promote the migration of T cells to the tumor microenvironment, resulting in the poor homing ability of CAR-T cells to the tumor site after intravenous infusion.

#### 2.1.3. Local Drug Delivery

When treating blood tumors, CAR-T is usually infused intravenously. Unlike most blood tumor targets, which are single and specific, there are few tumor-specific antigens (TSA) in solid tumors. At present, most of the highly expressed antigens in tumors are tumor-associated antigens (TAA), which are also expressed in normal tissues.

This brings a high risk of the off target, and the homing ability of intravenous infusion of CAR-T cells is poor [16]. Different from the scattered blood tumor cells, solid tumors often form solid masses, which, together with abundant tumor-related fibroblasts (CAF) and blood vessels, form a natural physical barrier. Therefore, in solid tumors, local administration can be used to improve the problem of CAR-T infiltration and transport. At present, the local delivery combined with CAR-T therapy mainly includes intratumoral drug delivery, local implantation, and intracavitary drug delivery.

Firstly, the way of intratumoral administration. HER2 is not expressed in normal central nervous system tissues but expressed in a variety of central nervous system tumors, so it has become an attractive target for the treatment of central nervous system tumors [51,52]. Nicholas A. Vitanza et al. [53] created three kinds of HER2 antigen-specific CAR-T cells with different lengths of spacers—short, medium, and long—and injected them into the glioma transplanted in situ in mice. Two weeks later, the intracranial tumors of the mice in the HER2-specific CAR-T group (M spacer HER2-specific CAR-T cells) injected with medium-length spacer completely disappeared, and the survival rate of the mice within 90 days reached 100%, which was significantly better than the other two kinds of spacer CAR-T groups and the control group [53]. In addition, intratumoral drug delivery has achieved good results in the study of pleural malignant mesothelioma, head and neck cancer, and other malignant tumors [54,55,56,57,58].

Secondly, in addition to local intratumoral injection, local implantation can also be used for drug delivery. Local implantation is of great significance for the treatment of tumors that are easy to recur or metastasize after surgical resection [59]. Because local drug preparation can directly locate the drug on the wound after tumor resection through a certain way of administration and improve the drug concentration at the focus, so as to effectively prevent tumor recurrence in situ or even distant metastasis with a smaller dose, it is expected to become a more ideal post-operative treatment plan. The hydrogel is widely concerned because of its good biocompatibility, degradability, ease of containing a variety of therapeutic agents, and controllable drug release [60]. Therefore, local combined immunotherapy through a biocompatible hydrogel repository after surgery may represent a treatment direction that can prevent tumor recurrence. Edikan A. Ogunanaike et al. developed a fibrin gel that can accommodate the load of CAR-T cells, promote their gradual release, and form in situ in the resection cavity after surgical removal of GBM tumor mass [61]. It is proven that compared with free CAR-T cells directly inoculated into the tumor resection cavity, local administration of CAR-T cells through fibrin gel is superior in controlling local tumor growth. In addition to the simple use of hydrogel to deliver CAR-T cells, the therapeutic effect can also be improved by adding cytokines. Gianpietro Dotti et al. loaded two different types of cells (CAR-T cells and anti-PD-L1 antibody-modified platelets) and nanoparticles containing IL-15 cytokines into an implantable hydrogel with good biocompatibility at the same time [62]. Controlled release of CAR-T cells was achieved through the slow-release ability of the hydrogel, to avoid a large inactivation of CAR-T cells caused by rapid release, and to maintain the activity of CAR-T cells through the role of IL-15. At the same time, in the inflammatory environment, platelets with PD-L1 antibodies modified on the surface can be activated and release anti-PD-L1 antibodies, block the inhibition pathway of the immune checkpoint, enhance the killing capacity of CAR-T cells, and more effectively inhibit the recurrence of cancer after surgery. However, it is necessary to optimize the delivery of CAR-T cells to the resection chamber after surgery. Effective local delivery of CAR-T cells must ensure wide coverage of the resection chamber surface to maximize the possibility of T cells contacting with residual tumor cells. Therefore, the selected scaffold needs to be able to form good contact with the tumor cavity [63]. In addition, the scaffold should ensure the integrity and function of these CAR-T cells. For example, the biocompatible fibrin gel has excellent biodegradability and wound-healing characteristics, which can degrade well without affecting wound healing [64]. Therefore, the selection of appropriate biological scaffold materials is of great significance to effectively play the anti-tumor role of CAR-T cell therapy.

Finally, in addition to intratumoral injection and local implantation, local drug administration in the sheath and other parts can also be used [65]. Kuniaki Tanaka et al. showed that intrathecal (IT) donor lymphocyte infusion can be used to treat ALL in the central nervous system without serious adverse events [47,66]. One patient received an intracranial infusion of IL13R α. Patients with glioblastoma treated with 2-CAR-T cells showed disease regression for at least 7.5 months [65]. This inspired us that direct infusion of CAR-T cells into the central nervous system could be a new way to treat ALL in the central nervous system. In general, to improve the therapeutic effect of CAR therapy in solid tumors, different drug-delivery methods can be adopted according to the characteristics of the tumor site.

### 2.2. In Vivo Production Mode

In addition to producing CAR series cells in vitro, CAR cells can also be directly produced in vivo [67,68,69,70]. Compared with the traditional preparation process of CAR-T, the advantages of producing CAR cells in vivo are that the arduous and time-consuming T-cell programming steps occur in vivo, eliminating the complex engineering of T cells in vitro, such as single collection and the preparation process of T cells transfected with viruses in vitro [71,72]. The process of preparing CAR-T cells is much simpler, which can greatly reduce the treatment cost. In vivo CAR-T treatment process, specifically, through a single, universally applicable drug product, the system delivery vector encoding CAR will be directly used to transduce patients’ T cells and can decode new genes and generate CAR within 1 or 2 days. This method can not only reduce the treatment cycle for patients but can also save time and is relatively simple. At the same time, there is no need to use initial cells, and there is no problem with the cell source, which can avoid graft versus host reaction, improve the safety of treatment, and make CAR-T in vivo a real universal CAR-T treatment method. In this process, the CAR coding system is directly used to transduce the T cells of patients. At present, the in vivo method has not entered the clinic, mainly because of the lack of an appropriate carrier platform [73]. Therefore, it is particularly important to select a safe and effective carrier platform. Next, we will introduce several vectors to generate CAR-T cells in vivo, including viral vectors (lentivirus, adenovirus-related vectors) and nonviral vectors (transposons, nanoparticles, etc.).

#### 2.2.1. Viral Vector

Viral transduction is the most common method of gene transfer. At present, the method of preparing CAR-T cells using viruses as vectors also dominates. Some studies show that retroviral or lentiviral gene transfer can lead to high transduction efficiency (4% to 70%) [74]. On the route of CAR-T transduction directly in vivo using viral constructs, the first thing to face is the safety of putting the in vitro transfection step into the body. In addition, the high selectivity of receptor-targeted gene-delivery vectors and target cells is also an improvement direction. The following table summarizes the details of marketed CAR-T drugs prepared using viral vectors (Table 1). Currently, there are no therapeutic drugs that directly generate CAR-T cells in vivo on the market.

##### Lentivirus

At present, CAR-T cells produced in vitro are mainly prepared by viral vectors, including lentiviruses or retroviruses. Among them, lentivirus vectors (LVs) show special prospects as gene therapy tools due to their efficient gene integration and stable expression in target cells [82]. However, for the treatment of CAR in vivo, the safety of using viral vectors needs to be concerned. Now there are also ways to generate functional CAR-T cells directly in the body through viral vector transmission for cancer immunotherapy [67,69,83]. However, for the treatment of CAR in vivo, the safety of using viral vectors needs to be concerned. The use of T cell-targeted virus vectors is expected to prevent CAR from being delivered to non-T cells and avoid fatal consequences to patients. Receptor targeting is a reasonable vector design method [73]. The carrier has selectivity to specific cell surface receptors, such as immune cell markers such as CD3, CD4, CD5, or CD8, and selectivity is achieved at the level of cell entry. It is a promising research direction to improve these vectors so that they preferentially transduce cells displaying target receptors on their surfaces, with the potential of lower toxicity and higher efficacy.

To eliminate the complex steps of producing CAR-T therapy in vitro and using chemotherapy to eliminate T cells in patients, EXUMA Company proposed a strategy to use a lentivirus vector that can self-inactivate to express and construct transgenic CAR-T cells, deliver the lentivirus vector into the body through subcutaneous injection, and transfect the lentivirus vector into CAR-T cells in vivo [84]. This vector is a self-inactivating lentivirus vector carrying CAR-encoding CD19. Its capsid protein is modified to target and activate CD3-positive T cells. In this study, the lentivirus vector successfully generated CD3-positive CAR-T cells targeting CD19 in mice. At the same time, B cells in mice continued to decline, and B cells were completely cleared in some mice, indicating that lentivirus vector delivered to infect CD3-positive T cells in vivo can generate CAR-T cells with normal functions, and can produce anti-tumor immune response against B cell malignant tumors. The company’s research provides new opportunities to overcome the challenges of cell therapy in terms of production time, scale, and cost.

Umoja Biopharma is also studying UV-VV100, which includes autologous T cells that carry out genetic engineering in vivo through lentivirus transduction [85]. Using the Vivo Vec drug delivery platform, it co-expresses CD19-specific chimeric antigen receptor (CAR) and a new rapamycin-activated cytokine receptor (RACR), realizing the rapid preparation and control of CAR-T cells, and is used for potential treatment of malignant blood tumors with cd19-positive b cells. In June 2021, the therapy will be listed as a pre-clinical development. This also provides some guidance for the subsequent research on the generation of CAR-T therapeutic cells in vivo by viral vector delivery [68]. Alexander Michels et al. studied the availability of mCD8-LV, making it possible to generate an isogenic mouse model of CAR-T cells in B-cell lymphoma in vivo [86]. Mustang announced cooperation with the Mayo Clinic to promote the development of in vivo genetic engineering CAR-T by using the new technology of Mayo Clinic. First, peptide drugs were used to drive the proliferation of T cells in patients, and then viral constructs were injected into patients’ lymph nodes to generate CAR-T cells [87]. Although the integration characteristics of lv can stably express genes in rapidly dividing cells (such as activated CAR-T cells), there is only limited control on the pharmacokinetics of transduction after in vivo administration. Compared with lv, AAV has a good level of biological security, as well as the safety of gene transmission in vivo.

##### Adeno-Associated Virus

Adeno-associated virus (AAV) is a small non-enveloped virus. The immunogenicity of AAV is very low, and it has almost no pathogenicity in humans [88]. This also means that the use of AAV vectors will not cause serious immune reactions, and the virus structure it retains does not have pathogenicity, which greatly improves the safety of AAV vectors. AAV consists of a non-envelope protein capsid containing a single-stranded DNA genome. Its genetic modification is usually transient, especially in proliferating cells (such as activated lymphocytes). The effectiveness of the recombinant AAV vector largely depends on the interaction between the viral capsid and the receptor molecules on the surface of the target cell. After interaction with their target receptors, clathrin mediates endocytosis and intracellular transport (the details of these processes are determined by the serotype of the capsid). AAV enters the nucleus and releases single-chain transgenes.

At present, it seems to be a simple strategy for AAV-mediated CAR gene transfer that the AAV vector targets the surface marker of mouse T lymphocytes. Receptor targeting depends on surface engineering, mainly including two modifications: elimination of natural receptor binding and display of targeted ligand. The anchor repeat protein (DARPins—CD8a chain recognition) designed by Alexander Michels et al. shows high affinity for the target receptor and is beneficial to the modification of AAV capsid. mCD8-AAV has high gene transfer activity [86].

Given that the integration mode of the above viral vectors significantly reduces the risk of carcinogenic transformation and random transgenic integration, it is feasible to use this vector to produce safe and efficient CAR-T cell products, but the manufacturing process of viral vectors is complex, costly, and strictly controlled [89]. In terms of regulation, it is difficult to comply with all aspects of cGMP at the early stage of clinical trials [90]. In addition, there are also challenges in the evaluation of vector efficacy and safety as well as in the transportation of virus vectors.

Even though most gene therapies are based on viral vectors at present, in recent years, emerging non-viral vectors are considered safer alternatives to viral vectors because of their high safety and low cost, and non-viral research is showing a rising trend [91]. In the research of CAR-T therapy, the application of non-viral vectors is also an important direction.

#### 2.2.2. Nonviral Vector

Compared with CAR-T cells prepared by viral vector systems, CAR-T cells prepared by non-viral vectors have advantages in many aspects [92,93,94]. The production process of a non-viral vector system is simpler and easier to conduct quality control. At the same time, in terms of storage, the stability of plasmid DNA used by non-viral vectors is higher. In this link of transfection, the cost of a non-viral vector is only 1/10 of that of the viral vector. CAR-T cells transfected by non-viral vector systems will also become a sharp sword for us to treat solid tumors. Because one of the difficulties in treating solid tumors is that it is difficult to find targets that distinguish solid-tumor cells from normal cells. If CAR-T cells are prepared with non-viral vectors, they can be transfected with multiple genes, and multiple targets can be used to locate solid-tumor cells to achieve more accurate killing. It is possible to improve the efficiency of CAR-T cells in the treatment of solid tumors. Non-virus vectors are mainly represented by transposons. Nano-carriers such as LNP can also be used for CAR delivery in vivo. The following table summarizes the details of some studies using non-viral vectors to achieve CAR therapy (Table 2). Here, we mainly introduce the use of transposons, lipid nanoparticles, and other delivery technologies in the treatment of CAR in vivo.

##### Transposon

The transposable subsystem is a non-viral gene-delivery system that has been gradually applied in the field of cell therapy in recent years [102]. Transposons have been proven to be able to achieve safer gene integration while maintaining efficient delivery efficiency, with limited miss effect, and can generate stable and reliable transgenic cells with gene expression. Common transposition subsystems include Sleeping Beauty (SB), PiggyBac, etc. Compared with viral gene delivery, transposon-based non-viral delivery has many advantages. In terms of production and manufacturing, it is easy to cultivate and its cost is low. The purification process is simple, and it takes a short time. In terms of delivery capacity, it has broken through the capacity limit of virus vectors and can deliver a large number of genetic materials. In terms of safety, low immunogenicity and low preference of insertion sites reduce the risk of adverse reactions and mutagenicity.

The goal of non-viral vector gene transfer based on transposon technology is to change the mode of gene transfer and eliminate the need to culture (T) cells in vitro in cell therapy. It can also accelerate treatment and reduce costs. The Sleeping Beauty (SB, “Sleeping Beauty”) transposon system is a member of the Tc1/mariner transposon superfamily. After Ivics and others revealed its transposon activity in 1997, it was gradually developed as a non-viral vector [103]. It is another non-viral gene-transfection method besides the preparation of CAR-T by lentivirus and retrovirus. Precigen, a biopharmaceutical company focusing on the development of innovative gene and cell therapy, announced that based on its non-viral vector UltraCAR-T™. The cell therapy PRGN-3006 of the treatment platform has obtained the “orphan drug” qualification (ODD) from the U.S. FDA for the treatment of recurrent refractory acute myeloid leukemia (AML) [104]. This study utilizes UltraCAR-T™. The platform’s advanced non-viral delivery system co-expresses chimeric antigen receptors, membrane-bound interleukin-15 (mbIL15), and a switch that targets indications and achieves precise control. The co-expression of mbIL15 can enhance the specificity and persistence of UltraCAR-T cell immune activation, help solve the problem of T-cell failure, and control the switch to control T cells and improve potential safety. The advantage of this research method is that UltraCAR-T cells can be rapidly produced using proprietary non-viral gene-transfer technology without proliferation in vitro, thus greatly reducing the waiting time of patients. SB system is likely to replace the current cumbersome gene knockout and RNA interference technology in the future and play an important role in functional genomics research and gene therapy. The two highlights of this technology are that it is operated in a non-viral manner, which improves safety and can prepare CAR-T cells very quickly. In addition, the FDA approved Precigen’s PRGN-3006 application for a new investigational drug (IND). PRGN-3006 UltraCAR-T is a multi-gene CAR-T cell therapy, which uses Sleeping Beauty Platform (SB system) to co-express chimeric antigen receptor, membrane-bound interleukin-15 (mbIL-15), and control switch for recurrent or refractory AML and higher-risk MDS. It will be used to treat patients with recurrent or refractory acute myeloid leukemia (AML) or high-risk myelodysplastic syndrome (MDS). Ziopharm Oncology, an American biotech company, also pioneered a non-viral method for manufacturing CAR-T cell therapy based on the SB transposition system, to eliminate the need to culture T cells in vitro.

The non-viral vector system based on transposon technology, in addition to the SB transposon mentioned, also includes the piggyBac transposon [105]. The PB transposon was first found in the moth in 1983 and was successfully used in gene manipulation of mammalian cells in 2005. They can overcome the problem of transient expression and have the inherent ability of stable expression and integration into nuclear genes. In recent years, great progress has been made in the application of CAR-T cell therapy [106]. The non-virus piggyBac platform does not use virus vectors. The most important thing about this technology is that the transposase can remove transposons in a completely seamless manner without leaving any sequence or mutation. It can effectively transfer a large number of CAR molecular genes to T cells and has the advantages of a high percentage of Tscm produced, large load (more than 200 kb has been proved), low risk, stability and efficiency, low production cost, and short time. P-BCMA-101 includes stem cell memory T cells that target B-cell mature antigen (BCMA), can self-renew, and has a long life. It is an autogenous CAR-T therapy developed based on Poseida’s PiggyBac platform non-viral vector technology. The cost of this method is low, and a high percentage of stem cell memory T cells (Tscm) can be generated based on this method. Currently, it is in the phase 1 clinical development stage of treating recurrent/refractory multiple myeloma.

##### Lipid Nanoparticles

The lipid nanoparticles (LNP) system is a leading non-viral delivery system at present [107]. LNP vector has been widely used and verified in the COVID-19 mRNA vaccine, and it is also used in CAR therapy [108,109,110]. The LNP delivery technology is mainly used to generate CAR-T cells in vivo by delivering the CAR gene or mRNA in CAR therapy [111]. Nanoparticles are a suitable carrier for editing T cells in vivo, which can transfer the CAR gene structure to the T cells of patients, which can avoid the expensive and time-consuming steps of preparing CAR-T cells in vitro. It is a method that has been proven in mice [99]. In addition, nanoparticles can also deliver mRNA that can encode CAR into mice. Some studies have successfully achieved the generation of CAR-T cells in vivo and can repair the heart function of mice [112]. Here is some information about the direct delivery of the CAR gene and the mRNA encoding CAR. The process of LNP entering the cell is shown in the following Figure 2.

##### Delivering CAR Plasmid

Lipid nanoparticles can be used as tools to deliver genes for CAR therapy. CHEN et al. developed a nanoparticle containing the gene encoding the chimeric antigen receptor (CAR), which was loaded in the hydrogel for injection into the brain cavity left after the removal of glioblastoma [113]. Nanoparticles enter macrophages and target their CAR expression to a glycoprotein called CD133 (a marker of glioma stem cells). The research results showed that the macrophages and microglia around the resection cavity of glioma can express CAR macrophages (CAR-M) that can target glioma stem cells through genetic engineering and make them produce in vivo. This method can eradicate the residual glioblastoma cells, prevent GBM recurrence and drug resistance, improve the persistence of CAR-T cells in the tumor site, and significantly improve the survival rate. Therefore, this method may provide a more effective immunotherapy strategy for patients with recurrent malignant tumors and is worthy of further clinical verification. Stephan et al. [98,99,100] developed a biodegradable nanoparticle that can transmit the CAR coding gene, and used molecular markers to make it adhere to T cells like a burr. Once T cells engulf these particles, they enter the nucleus along the cell’s internal transport system and then dissolve. The CAR gene it carries will be integrated into the chromosome in the nucleus, and T cells can decode the new gene and generate CAR-T cells in a day or two. Then it can effectively kill the tumor, and its therapeutic effect is equivalent to that of CAR-T prepared by traditional lentivirus [99,114].

##### Delivering CAR mRNA

In addition to using nanoparticles to deliver CAR gene carriers, CAR-T cells generated in vivo can also directly deliver the mRNA encoding CAR plasmid. The treatment of CAR-T cells based on mRNA has been verified in many previous cancer studies, including ALL, melanoma, and Hodgkin’s lymphoma, and has been proven to be as effective as a stable expression of CAR-T cells in reducing short-term disease burden [115,116]. Although messenger RNA (mRNA) has been explored as a promising strategy to mitigate the adverse effects associated with viral vectors, it usually requires electroporation to deliver T-cell mRNA, which may have cytotoxicity. Therefore, it is particularly important to select appropriate vectors to deliver mRNA to T cells. RURIK J G et al. [95] designed to use lipid nanoparticles (LNP) to deliver the mRNA-encoding CAR gene. They modified the targeted antibody on the surface of the LNP, and LNP encapsulated the mRNA to transport it to specific T lymphocytes [95]. When LNP enters T cells, it will release mRNA and program T cells into functional CAR-T cells. CAR-T cells are endowed with specificity and can target fibroblast-activated protein (FAP) to attack activated fibroblasts. The same principle can also be applied to tumor cells by modifying LNP-targeted receptors. The research of Margaret M. Billingsley et al. screened the ionizable lipid components and formulations in LNPs and verified that LNP is a potential alternative strategy for MRA-based CAR-T cell engineering in vitro [97]. The LNP delivery method is similar to the mRNA vaccine, and it can generate CAR-T cells in the body with only one injection, which makes individualized immunotherapy an important step towards the direction of easy and cheap access, and is expected to solve the problems of the complex process, long cycle, and high price of CAR-T therapy. The following Figure 3 shows that lipid nanoparticles arrive at tumor sites in different ways of administration to express CAR to kill tumor cells.

In vivo production of CAR-T technology is important to the field of cancer treatment, as it simplifies the treatment process and can reduce production costs and may have better efficacy and lower side effects. It is expected that more patients will benefit from CAR-T cell therapy. As of 2024, some in vivo CAR-T generating programs, such as Interius’ INT2104 (NCT06539338) [117] and Umoja’s UB-VV111 (NCT06528301) [118], have already been approved for early human trials, but no efficacy data have yet been published. More Phase I/II trial data is expected to be disclosed in 2025–2030, but large-scale clinical applications will still take a long time to validate. The core obstacle of generating CAR-T cells in vivo lies in the balance between delivery efficiency and controllability, so it needs to be gradually broken through technological innovations such as precision targeting carriers. Moreover, due to the complexity of the in vivo environment, there is still a lack of reliable technological means to monitor the dynamics of CAR-T cells and accurately regulate CAR-T cells, so there is plenty of room for progress.

### 2.3. Synergistic Treatment

Tumor microenvironment (TME) is also one of the reasons that the efficacy of CAR therapy in solid tumors is affected [119]. In solid tumors, tumor microenvironment is a serious obstacle to CAR-T cells. First, there are some suppressive immune cells in the tumor microenvironment, such as regulatory T cells, tumor-associated macrophages, and tumor-associated neutrophils. These cells produce a variety of immunosuppressive factors and cytokines, such as TGF-b, PGE2, IL-10, and IL-4, as well as reactive oxygen species, which are conducive to the survival of tumor cells. Secondly, the inhibition of cytotoxic T lymphocytes to react with a large number of tumor-associated fibroblasts (CAFs) in tumor tissue has built a good environment for the development of tumors (accounting for more than 50% of stromal cells). CAFs play an important role in cancer. They can shape the tumor matrix, form a drug or therapeutic immune cell penetration barrier, and prevent the drug and immune cells from penetrating into the tumor tissue. In addition, CAFs (not only CAFs) can also inhibit the function of immune cells by secreting a variety of cytokines or metabolites, promote tumor development, invasion and metastasis, and thus reduce the effect of tumor treatment. Finally, there is a complex tumor microenvironment in solid tumors. The glycolytic metabolism of tumor cells makes the environment low in oxygen, acid, and nutrition, and is prone to oxidative stress. Immune cells may not be able to proliferate under the conditions of hypoxia, nutrient deprivation, lack of essential amino acids, and high concentration of metabolic acids [14], which is conducive to tumor progress. The complex tumor microenvironment (TME) plays an important role in tumor malignant progression, immune escape, and therapeutic resistance, which limits the efficacy of CAR-T cell therapy in clinical trials of cancer patients.

For the above-mentioned factors, we can intervene against non-tumor cells and microenvironment components, respectively. For example, for immunosuppressive cells, immune checkpoints can be used together to transform the immune response from tumor promoting to tumor inhibiting [120,121]. PD-1, LAG3, and CTLA-4 are common immune checkpoint molecules in TME [120,122]. CAR-T cells can provide infiltration for immunogenic silenced tumors, while immune checkpoint inhibitors can reverse the inhibition of CAR-T cells and restore the persistence of their functions, transforming the immune response from tumor-promoting type to tumor-inhibiting type [123]. Leonid Cherkassky et al. studied the interference of PD-1/PD-1 ligand [PD-L1] pathway, and restored the effective function of CD28CAR-T cells by blocking PD-1 antibody checkpoint, blocking PD-1 shRNA in cells, or PD-1 dominant negative receptor, suggesting that PD-1/PD-L1 blocking may be an effective strategy to improve the therapeutic efficacy of CAR-T cells [124].

For microenvironment components, cytokine therapy can be supplemented. It is expected to avoid adaptive drug resistance and significantly improve the prognosis and survival of patients with CAR therapy by combining with anti-tumor and multi-target immunotherapy drugs [125]. At present, some studies have modified CAR-T cells to over-express inflammatory cytokines (such as IL-12, IL-15, and IL-18) [126,127,128,129]. Such CAR-T cells are called “armored CAR-T cells”, which can regulate the local microenvironment. Interleukin 18 (IL-18)-secreting CAR T cells can modulate the tumor microenvironment as well as enhance an effective endogenous anti-tumor immune response, mainly because the enhanced killing is accompanied by the recruitment of endogenous anti-tumor immune effector cells (e.g., T-cells and macrophages), which achieves a reversal of the immunosuppressive microenvironment.

It has been confirmed that in ovarian cancer, IL-12 can improve the proliferation and survival of T cells, resist apoptosis, and PD-1-induced functional inhibition [130,131]. In addition, relevant studies on CAR T cells secreting IL-18 showed that these cells had increased proliferation and infiltration ability and could recruit endogenous immune cells to regulate the tumor microenvironment [132,133]. This approach can bypass the key points of tumor immunosuppression. Therefore, this kind of engineered local cytokine transport circuit may provide a potential universal strategy to drive the effective T cell activity of solid tumors that resist immunosuppression. This approach can bypass the key points of tumor immunosuppression. Therefore, this kind of engineered local cytokine transport circuit may provide a potential universal strategy to drive the effective T cell activity of solid tumors that resist immunosuppression.

### 2.4. Challenges and Solutions

In solid tumors, CAR-T therapy is often accompanied by some fatal complications, especially cytokine syndrome (CRS) and CAR-T-related encephalopathy syndrome (CRES) [134]. Severe toxicity is still the main challenge of CAR-T cell therapy. Therefore, it is of great significance to establish a systematic strategy to effectively avoid the risk of serious side effects, to realize accurate prognosis prediction through dynamic monitoring of the treatment process, and to construct a multi-dimensional adverse reaction management system for CAR-T cell therapy to enhance the safety of the treatment and guarantee the clinical efficacy. In addition, in terms of the selection of delivery materials, there are also challenges in this area due to the different characteristics of materials. Therefore, searching for suitable delivery materials (such as hyaluronic acid and alginate) is a direction for improving the efficacy of CAR-T therapy in the future [135,136,137].

#### 2.4.1. Cytotoxicity

##### Cytokine Release Syndrome (CRS)

In the treatment of hematological malignancies, the activation and expansion of CAR T cells can lead to a large amount of cytokine/chemokine secretion, leading to CRS toxicity. The first symptom of CRS is fever [138]. The patient can rapidly develop into hypotension and hypoxemia, and the function of important organs is damaged. The study found 24 kinds of cytokines and soluble cytokine receptors (including IFN γ). The higher the peak value of IL-6 and GM-CSF, the more severe CRS, especially IL-6, indicating that inflammatory factors play an important role in the pathogenesis of CRS. Tocilizumab, an IL-6 receptor-blocking antibody, has been widely used in the treatment of severe CRS with obvious effects [139,140]. Glucocorticoids are also used to treat severe CRS but may harm CAR-T cell function [141]. The factors leading to severe CRS include the high load of malignant cells, a large dose of CAR-T cells, and the selection of the costimulatory signal domain of the CAR structure. Understanding these factors enables us to adopt a risk-adaptive approach, including reducing the dose of CAR-T cells. The humanized mouse CRS model showed that activated CAR-T cells induced a large number of monocytes/macrophages to aggregate and secrete large amounts of IL-6, IL-1, and nitric oxide. IL-1 β may play an important role in the pathogenesis of CRS, so IL-1 can be used in β Inhibitor treatment [96]. Biomarkers predictive of CRS such as ferritin, because of their correlation with the severity of CRS, allow for standardized CRS grading and toxicity management using multi-institutional scoring systems. Patients with high-grade CRS can be salvageable with early and aggressive management, and monitoring patients after successful CRS treatment is an important tool for managing adverse events [134,142].

##### Off-Target Effects

The recognition of “tumor outside the target” generated by CAR-Tcells directly attacking normal tissues is the most significant toxicity. One of the challenges of CAR therapy targeting solid tumor antigens is the off-target effect. Solid tumor antigen is usually expressed at different levels in normal tissues. Therefore, antigen selection is crucial in CAR design, which can not only ensure the therapeutic effect but also limit the off-target toxicity. A potential way to overcome the targeting of solid-tumor antigens that also exist in normal tissues is to target only the post-translational modification of tumors, such as truncated O—glycans overexpressed in solid tumors, such as Tn (GalNAca1-O-Ser/Thr) and sialyl Tn (STn) (NeuAca2—6-GalNAca1-O-Ser/Thr). Four major CAR-T cell targets have been studied, including TAG7228, B7-H3, MUC1, and MUC16 [16,143,144]. To expand the clinical application of CAR-T cell therapy in blood malignancies and solid tumors, further develop innovative strategies to reduce antigen escape and select antigens that can induce sufficient anti-tumor efficacy, while reducing toxic and side effects. Targeting off-target toxicity requires improving the specificity of CAR-T cells by optimizing the structure of the CAR. The core countermeasure strategy for off-target toxicity of CAR therapies focuses on the precise screening and validation of tumor-specific antigens to increase specific recognition of target proteins, thus circumventing off-target toxicity at the source [142].

##### Neurotoxicity

Neurotoxicity is the second major complication of CAR-T cell therapy, which can be fatal. Neurotoxicity, or CAR-T cell-associated encephalopathy syndrome, is typically characterized by confusion, delirium, expressive aphasia, a certain degree of myoclonus, and occasional brain edema and epilepsy [145]. Although the exact pathophysiology of neurological side effects is still unclear, whether the pathophysiology of neurotoxicity comes from the direct attack of CAR-T cells on central nervous system tissues or from extensive cytokine-mediated inflammation has not been studied. However, it is believed that neurotoxicity is the result of increased cytokine levels. In addition, some studies have shown that although there is a lack of CD19 expression in the affected brain regions, there is a certain correlation between neurotoxicity and the presence of CAR-T cells in cerebrospinal fluid. However, CAR-T cells were not detected in the spinal fluid of all neurotoxic patients. In the research of D. Posey, Jr. [146] by analyzing single-cell RNA sequencing data in human brain samples, the research team found that CD19 is expressed on human brain wall cells, which may be the cause of neurotoxicity caused by CD19 CAR-T cell therapy. The therapeutic effect of Tocilizumab monoclonal antibody on neurotoxicity is limited, which may be due to the difficulty of the antibody entering the central nervous system. Therefore, large doses of glucocorticoids are currently used to treat neurotoxicity [147]. CAR-T-associated neurotoxicity was initially graded according to the CTCAE, and the CRES grading system was subsequently introduced for management. Similar to CRS, neurotoxicity is reversible in most patients, but resolution of neurologic symptoms usually takes longer than with CRS [148,149].

In summary, toxicity and adverse reactions to CAR-T therapy are a serious risk factor and are critical to patient safety. The management of CAR-T-cell-related toxicity is an evolving field as clinical experience grows, and strategies for the management of adverse reactions emphasize prevention, continuous monitoring, rapid detection, and symptomatic treatment for intervention [142,149].

#### 2.4.2. Delivery of Materials

In this paper, we summarized some vectors used to deliver the CAR gene in vivo, namely viral vector and non-viral vector. The viral vector is one of the earliest biological drug-delivery technologies developed. The emergence of viral vectors was mainly to solve the problem of DNA delivery. Lentivirus is the most mature viral vector type in an industrial application at present and plays a key role in immune cell therapy, such as CAR-T therapy. Compared with other types of retroviruses, lentiviruses can penetrate the nuclear membrane, infect a wider range of cell stages, and effectively infect both dividing and non-dividing cells. Most retroviruses cannot penetrate the nuclear membrane and can only wait for mitosis to enter the nucleus. Therefore, the lentivirus vector has gradually replaced the original retroviral vector system in development, but this does not mean that the lentivirus system is safe. Lentivirus inherits a little bit from retrovirus but also inherits the instability of its random insertion.

With the approval of several gene therapy products, adeno-associated virus (AAV) has been repeatedly mentioned as a new viral vector delivery technology. The safety and clinical value of AAV is also demonstrated in the clinical use of these products. The immunogenicity of AAV is very low, and there is almost no pathogenicity in humans. This means that the use of an AAV vector will not cause a serious immune response, and the virus structure retained by the AAV vector is not pathogenic, which greatly improves the safety of the AAV vector. However, clinical LV and others have strict requirements on relevant production and testing technologies, which further increases the cost of CAR-T cell therapy and limits its expansion in clinical applications. Therefore, non-viral vector-dependent transfection is also gradually emerging, providing more options for the preparation of CAR-T cells. Non-viral vectors (especially LNP, dozens of related clinical experiments are underway) have recently received worldwide attention, and research on their use in gene therapy is also developing. The following table summarizes some of the advantages and disadvantages of using different carrier delivery CARs (Table 3).

#### 2.4.3. Technical Means

The challenges in the technical means of CAR therapy mainly focus on (1) Mass production: In the process of CAR-T treatment, it is necessary to culture the technically modified T cells in vitro, reach the number of cells that meet the treatment requirements, and then infuse them back into the patient’s body to kill cancer cells. In the production of CAR-T, it is a common practice to supplement cytokines to enhance the proliferation of CAR-T cells in vitro. IL-2 can induce T-cell expansion and increase the number of early memory cells by regulating the concentration of IL-2. At present, the rapid expansion of T cells in vitro is still a challenge. Even with the participation of cytokines, the expansion speed and stability still need further exploration and optimization to achieve. (2) Rapid characterization and sterility before intravenous reinfusion: CAR-T cells need to undergo strict release testing before being reinfused into the patient’s body, mainly including biochemical testing, microbiological testing, and cytological testing. The quality of CAR-T cells directly affects the clinical treatment effect and the patient’s health. If not strictly tested, it may even endanger the patient’s life. (3) At present, the preparation of CAR-T cells in most enterprises is still purely manual production, which not only increases the risk of product contamination but also easily leads to product batch differences. Comprehensive process research and process validation of continuous production of multiple batches (at least three batches) should be carried out in the whole process of CAR-T cell product preparation.

## 3. Summary and Outlook

Although CAR-T treatment still faces various challenges, CAR-T cell treatment will completely change cancer treatment. With the progress of research, there is a great room to improve the safety and effectiveness of CAR-T cells in the treatment of solid-tumor patients and to obtain such treatment. In addition to the delivery strategy to improve the efficacy mentioned above, there are other CAR-T cell optimization strategies. For example, the strategy of constructing dual-target CAR in series or parallel to overcome the escape of antigen, Knock out CD33, etc., in targeted blood tumor treatment, or use gene mechanism and other designs in solid tumor treatment to solve the non-target Off-Tumor toxicity. We hope that CAR-T cells may prove to be a safe and feasible method for treating various malignant tumor patients.

In the future, the preparation of in vivo CAR-T cells may be more focused on improving the transfection efficiency, target specificity, and safety of gene delivery vectors, and appropriate technical approaches need to be selected to more precisely control the in vivo behavior of CAR-T cells, as well as to study how and how often CAR-T cells should be activated/inactivated in vivo to achieve maximum tumor-killing effect [157]. In vivo CAR-T technology is facing non-standardized production processes, imperfect quality control systems, and bottlenecks in large-scale production, resulting in cost advantages that have not yet been fully released [114]. Only through breakthroughs in dynamic tracking technologies (e.g., real-time imaging of multiple genomes), the establishment of automated production platforms, and the optimization of the delivery system can the theoretical low cost be converted into actual clinical benefits [157,158].

## Figures and Tables

**Figure 1 ijms-26-03206-f001:**
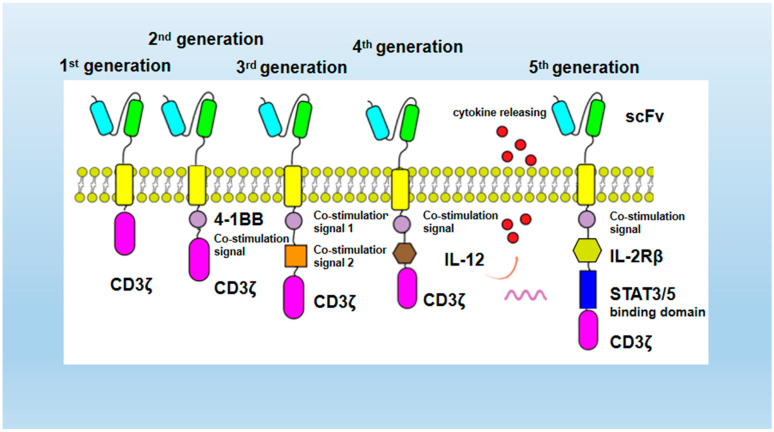
Schematic representation of the optimized iterations of the chimeric antigen receptor structure for generations one to five.

**Figure 2 ijms-26-03206-f002:**
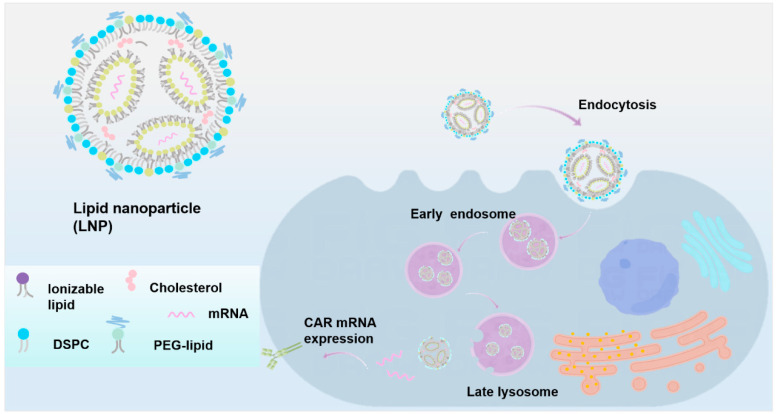
Intracellular process of LNP as a carrier for transporting CAR cargo. Images were created with Figdraw.com. The image elements were acquired on 5 December 2023.

**Figure 3 ijms-26-03206-f003:**
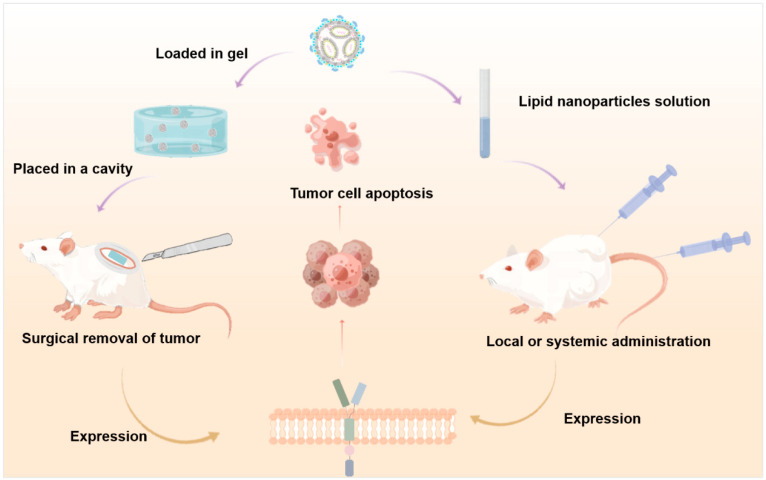
LNP expresses CAR at tumor sites through different drug-delivery pathways to achieve the goal of killing tumors. Images were created with Figdraw.com.

**Table 1 ijms-26-03206-t001:** Summary table of CAR-T products available successfully prepared in vitro.

Generation Mode	Company	Name	Indication	Antigen	Mode of Administration	Delivery Vector	Dose Range	Reference
In vitro	Novartis	Kymriah	RRFL	CD19	Intravenous	lentivirus vector	For patients below 50 kg: 0.2~5.0 per kg body weight × 10^6^ CAR-positive survival T cells For patients above 50 kg: 0.1 to 2.5 × 10^8^ CAR-positive survival T cells	[75,76]
	Gilead	Yescarta	R/R LBCL	CD19	Intravenous	retroviral vector	2 × 10^6^ CAR-positive living cells/kg body weight	[77]
	Gilead	Tecartus	R/R B-ALL	CD19	Intravenous	retroviral vector	2 × 10^6^ CAR-positive living cells/kg body weight	[78]
	Bristol Myers Squibb	Breyanzi	R/R LBCL	CD19	Intravenous	lentivirus vector	1.5 × 10^6^ to 70 × 10^6^ CAR-positive live T cells/mL	[79]
	Bristol Myers Squibb	Abecma	R/R MM	BCMA	Intravenous	lentivirus vector	300 to 460 × 10^6^ CAR-positive T cells	[80]
	Legend Biotech	Carvykti	R/R MM	BCMA	Intravenous	lentivirus vector	1.0 × 10^6^ CAR positive active T cells/kg body weight	[81]

RRMM: Patients with primary or refractory multiple myeloma; RRFL: Patients with recurrent or refractory (r/r) follicular lymphoma; R/R LBCL: Patients with recurrent or refractory large B-cell lymphoma; R/R B-ALL: Patients with Recurrent or refractory B-cell acute lymphocytic leukemia.

**Table 2 ijms-26-03206-t002:** Some currently under study use lipid nanoparticles or other carriers for the delivery of CAR genes.

Name/Identifier	Delivery Vector	Delivery Content	Target Head	Delivery Method	Size	Zeta Potential	Disease	Reference
CD5/LNP FAPCAR	Lipid nanoparticle	FAP-CAR mRNA	CD5	Intravenousinjection	80 nm	——	Cardiac fibrosis	[95]
AntiCD3-LNP/CAR19 + shIL6	Lipid nanoparticle	shRNA and the CAR gene	CD3	Intravenous injection	200 ± 16.5 nm	1.6 ± 0.2 mV	acute lymphoblastic leukemia	[96]
Ionizable Lipid Nanoparticle C14−4LNPs	C14−4LNPs	CD19 CAR mRNA	CD19	——	70.17 ± 0.41 nm	——	——	[97]
Foxo13A-NP	Engineered NPs	Foxo13A19–41BB mRNA	CD19	Intravenous injection	109.6 ± 26.6 nm	1.1 ± 5.3 mV	lymphadenoma	[98]
Poly(B-amino) ester polymer (PBAE) NP	Engineered NPs	194-1BBz CAR plasmid DNA	CD19	Intravenous injection	155 ± 40 nm	−7.8 ± 2.1 mV	leukemia	[99]
IVT mRNA nanoparticles	PBAE-447 polymer nanocarrier	CAR mRNA	CD8	Intravenous injection	106.9 ± 7.2 nm.	4 ± 2 mV	leukemia	[100]
comb-shaped pHEMA-g-pDMAEMA polymer	polymers nanocarrier	plasmidDNA/mRNA	——	——	100–150 nm	25–30 mV	——	[101]

——: The graphic indicates that the data is not disclosed.

**Table 3 ijms-26-03206-t003:** Summary of advantages and disadvantages of delivering CAR carriers in vivo.

	Advantages	Disadvantages	Reference
Viral vector	Rapid preparation	High cost	[150,151,152]
	Stable integration	Low security	
	Mature Technology Pathways	Immunogenicity	
Transposon carrier	Large gene fragments	Low transfection efficiency	[92,153,154]
	High biosafety	Complex plasmid design	
LNP	High RNA stability	Transient expression	[150,155,156]
	High delivery efficiency	Low transfection efficiency	

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
