# Peer review of "Strategies for Altering Delivery Technologies to Optimize CAR Therapy"

_ijms, 2025, doi:10.3390/ijms26073206_

Round 1
Reviewer 1 Report
Comments and Suggestions for Authors
In this work, the authors provide a detailed and well-organized review of the strategies to enhance the efficacy of CAR-T cells for broader and more effective therapeutic applications. Although the topic is promising and the review comprehensive, substantial revisions are needed to enhance clarity and depth. After careful reading, it is apparent that major revisions should be completed prior to publication.
- Would including a schematic representation of the CAR structure and its generational advancements enhance understanding? And should the clinical significance and therapeutic efficacy of each generation of CAR-T cells be described in more detail?
- In Table 3, I think the describe of the advantages and disadvantages of delivering CAR carriers in vivo is too short, it’s better to add more details and references.
- In the last part of the summary and outlook, I think it would be better to give more discussion of outlook and perspective as well as future directions, such as new emerging technologies to help delivery.
Author Response
请参阅附件。

Reviewer 2 Report
Comments and Suggestions for Authors
The core point of this review is that although CAR-T cell therapy has been successful in hematological malignancies, its application in solid tumors still faces challenges such as low delivery efficiency, immunosuppressive microenvironment, and high manufacturing costs. This review explores strategies to optimize CAR-T delivery technology (in vitro and in vivo production) and enhance efficacy to improve the wide applicability and safety of CAR-T therapy.
This review mainly summarizes the following points:
1. In vivo production of CAR-T cells: reducing manufacturing costs and simplifying treatment processes
2. Delivery technology optimization: enhancing the penetration and survival of CAR-T cells in solid tumors (local delivery strategy)
3. CAR-T structure optimization: improving efficacy and safety (discussing the fifth-generation CAR-T and Armored CAR-T respectively)
However, there are still some minor shortcomings in this article that can be discussed.
1. Although in vivo production of CAR-T cells is an important innovation to reduce costs, there is currently a lack of human trial data, and safety and efficacy are still unclear.
2. The analysis of the challenges of CAR-T treatment of solid tumors is not in-depth enough. Although the article discusses local delivery strategies, it does not analyze in detail how to overcome tumor microenvironment inhibition (such as the effects of TGF-β, IL-10).
3. The article proposes methods to reduce the production cost of CAR-T, but does not provide specific cost comparison data, such as the actual cost difference between in vivo delivery and traditional manufacturing. The lack of economic feasibility analysis makes the practical application prospects of this strategy unclear.
4. Although CAR-T side effects (such as CRS, CRES) are mentioned, there is no in-depth discussion on how to predict and manage these adverse reactions.
Round 2
Reviewer 1 Report
Comments and Suggestions for Authors
The authors addressed all my concerned and I think it should be published.